# Dynamic Change of R-Loop Implicates in the Regulation of Zygotic Genome Activation in Mouse

**DOI:** 10.3390/ijms232214345

**Published:** 2022-11-18

**Authors:** Hyeonji Lee, Seong-Yeob You, Dong Wook Han, Hyeonwoo La, Chanhyeok Park, Seonho Yoo, Kiye Kang, Min-Hee Kang, Youngsok Choi, Kwonho Hong

**Affiliations:** 1Department of Stem Cell and Regenerative Biotechnology, Institute of Advanced Regenerative Science, Konkuk University, Seoul 05029, Republic of Korea; 2Guangdong Provincial Key Laboratory of Large Animal Models for Biomedicine, Wuyi University, Jiangmen 529020, China

**Keywords:** R-loop, zygotic genome activation, transcription, DNA replication

## Abstract

In mice, zygotic genome activation (ZGA) occurs in two steps: minor ZGA at the one–cell stage and major ZGA at the two–cell stage. Regarding the regulation of gene transcription, minor ZGA is known to have unique features, including a transcriptionally permissive state of chromatin and insufficient splicing processes. The molecular characteristics may originate from extremely open chromatin states in the one–cell stage zygotes, yet the precise underlying mechanism has not been well studied. Recently, the R-loop, a triple–stranded nucleic acid structure of the DNA/RNA hybrid, has been implicated in gene transcription and DNA replication. Therefore, in the present study, we examined the changes in R-loop dynamics during mouse zygotic development, and its roles in zygotic transcription or DNA replication. Our analysis revealed that R-loops persist in the genome of metaphase II oocytes and preimplantation embryos from the zygote to the blastocyst stage. In particular, zygotic R-loop levels dynamically change as development proceeds, showing that R-loop levels decrease as pronucleus maturation occurs. Mechanistically, R-loop dynamics are likely linked to ZGA, as inhibition of either DNA replication or transcription at the time of minor ZGA decreases R-loop levels in the pronuclei of zygotes. However, the induction of DNA damage by treatment with anticancer agents, including cisplatin or doxorubicin, does not elicit genome-wide changes in zygotic R-loop levels. Therefore, our study suggests that R-loop formation is mechanistically associated with the regulation of mouse ZGA, especially minor ZGA, by modulating gene transcription and DNA replication.

## 1. Introduction

After fertilization, zygotes proceed in two steps of zygotic genome activation (ZGA), namely minor ZGA [1,2,3] and major ZGA, which ensures the proper development of preimplantation embryos [2,4,5,6,7]. Transcripts synthesized in growing oocytes are relatively stable and stored as maternal transcripts [8,9,10]. However, maternal transcripts need to be replaced by transcripts newly synthesized from the zygotic genome, called maternal–to–zygotic transition (MZT), after fertilization [9,11,12]. The timing and detailed mechanisms of ZGAs differ in a species-dependent manner [13,14,15]. In mice, minor ZGA occurs in the mid–to–late stage of one-cell embryos, whereas major ZGA occurs in two-cell embryos. It has been shown that both ZGAs are mechanistically interconnected in the regulation of transcription, especially by RNA polymerase II (Pol II) [16]. In addition, active translation of maternal mRNAs is required for MZT in mice [17]. Minor ZGA, characterized by the simultaneous occurrence of DNA replication and gene transcription, was shown to be essential for preimplantation embryo development [2]. Abe et al. showed that inhibition of Pol II elongation by treatment with 5,6-dichloro-1-β-D-ribofuranosyl-benzimidazole (DRB) from 4 h post–insemination (hpi) to 20 hpi led to aberrant minor ZGA in mice [2]. Most DRB-treated zygotes were arrested at the two-cell stage, and the two-cell stage embryos displayed characteristics of minor ZGA, but not major ZGA, even after being released from DRB [2]. In general, the transcriptional machinery is considered to be one of the barriers encountered by DNA replication folks, called transcription–replication conflict (TRC), which can cause genome instability [18,19,20]. To date, the precise mechanism by which the simultaneous occurrence of transcription and DNA replication is controlled during minor ZGA is largely unknown.

Recent studies have shown that the formation and resolution of the R-loop, a triple-stranded nucleic acid structure of the DNA/RNA hybrid, play essential roles in TRCs [20]. It has been shown that R-loop homeostasis is dynamically regulated and is potentially associated with MZT during early *Drosophila* embryogenesis [21]. In addition to its role in the regulation of replication and transcription, R-loops have been implicated in mitochondrial DNA replication, immunoglobulin class-switch, DNA double-strand break formation, DNA damage repair, and histone modification [20,21,22,23]. Although R-loops have been considered a rare structure in the past, recent evidence suggests that R-loops are widely spread throughout the whole genome and are involved in multiple molecular and cellular processes [22]. R-loops can be formed in the presence of 70% formamide in vitro and are thermodynamically more stable than DNA duplexes, in which the structure can persist even after formamide is removed [23]. In 1994, R-loop formation was confirmed in live bacteria in vivo [24]. Numerous studies have investigated the formation of R-loops and the underlying mechanisms by which R-loops are involved in biological processes.

Recent studies have reported that R-loops are approximately distributed in 5.8~10% of the human, yeast, and *Arabidopsis* genome [25,26,27], and are enriched at the transcription start site (TSS) and transcription termination site (TTS) [28]. The factors that control R-loop levels include DNA sequences, DNA topology, and proteins that regulate mRNA metabolism. R-loops are preferentially formed at sites with high levels of skewed GC content and sites with increased negative DNA supercoiling [29]. Malfunction of regulatory proteins has been shown to be associated with aberrant mRNA splicing and transport. For example, in eukaryotic cells, depletion of SRSF1 and THO/TREX increases R-loop levels [30,31,32]. There are two well-known factors that control R-loop levels. First, ribonucleases, RNase H1 and H2, degrade RNA in DNA/RNA hybrids. Second, helicases, senataxin and DDX5, reduce negative supercoiling. For example, human cells with senataxin silencing show increased R-loop levels [33,34,35]. Unlike the accumulated body of evidence of R-loop function in somatic cells, none of the R-loop functions in mouse preimplantation embryos have been elucidated.

In this study, we determined R-loop dynamics and found that R-loops persist throughout preimplantation embryo stages, even in metaphase II oocytes, and their levels decreased as pronucleus maturation proceeded in both pronuclei, in which the R-loop level was mostly weaker at the point of pronuclear stage 1 (PN1) to PN2 transition. We also provide evidence that R-loop dynamics are potentially associated with minor ZGA.

## 2. Results

### 2.1. Optimization of R-Loop Immunofluorescence (IF) in Mouse Preimplantation Embryos

To examine the dynamics of the R-loop in preimplantation embryos, the specificity of IF with the S9.6 antibody in mouse zygotes was examined. As shown in Figure 1, an IF method commonly used in other studies fails to detect clear S9.6 signals in the pronuclei; instead, “cloudy” S9.6 signals in the cytoplasm were detected. Therefore, we reasoned that an antigen retrieval step may be needed to expose the DNA/RNA hybrids to the S9.6 antibody in the pronuclei. Then, an antigen unmasking method with hydrochloric acid (HCl) that is routinely used for the detection of bromodeoxyuridine (BrdU), 5-methylcytosine (5mC), and its derivatives was adopted in our IF protocol [36,37]. The antigen retrieval step with HCl treatment is critical as S9.6 signals were barely detected with 2N HCl concentration, whereas S9.6 signals were clearly detected in both pronuclei with 4N HCl concentration (Figure 1A). Next, we asked whether the detected S9.6 signal is a true R-loop signal. To that end, 80 IU/mL of RNase H that specifically degrades DNA/RNA hybrids [38] was treated after the antigen retrieval with 4N HCl and neutralization with Tris-HCl (pH 8.5) steps. The S9.6 signals were evident in the pronuclei with the antigen retrieval and neutralization steps, and greatly diminished after Rnase H treatment. Therefore, the results suggest that antigen retrieval with HCl treatment significantly improves S9.6 IF for the detection of R-loops in mouse zygotes, and R-loops do exist in both maternal and paternal pronuclei (Figure 1B).

### 2.2. R-Loops Are Detected throughout Preimplantation Stage Embryos and All Cell Cycles

After optimization of R-loop IF, we sought to determine the dynamics of biogenesis and resolution of the R-loop during preimplantation embryonic development. Our IF analysis revealed that R-loops are detected as spotty patterns in the genome of pronuclei and persist from metaphase II stage oocytes to late preimplantation stage (blastocyst) embryos (Figure 2). It is interesting to note that the R-loop is detected in embryos that are in all stages of the cell cycle: G1, S, G2, metaphase, and telophase. The R-loop IF signals were also evident in both external and internal cells of the morula, and both trophoblast and inner cell mass of the blastocyst at similar levels. Therefore, R-loops are not rare nucleic acid structures and potentially participate in the preimplantation stage of embryo development (Figure 2).

### 2.3. R-Loop Level Is Dynamically Changed in One-Cell Embryo

Next, the R-loop dynamics throughout the PN1-5 of the one-cell embryos were analyzed in detail (Figure 3). The analysis revealed that R-loops were detected from the start of pronucleus formation to telophase at the end of the one-cell stage (Figure 2), and the relative mean intensity of the S9.6 signals was higher in the maternal pronucleus than in the paternal pronucleus throughout PN1-5 (Figure 3C). Interestingly, the R-loop levels decreased as pronucleus maturation proceeded in both pronuclei (Figure 3D), in which the R-loop level was mostly weaker at the point of PN1 to PN2 transition (Figure 3).

### 2.4. R-Loop Homeostasis Is Associated with Transcription and DNA Replication

Next, the effects of gene transcription and DNA replication on the R-loop metabolism were examined. To that end, inhibitors of transcription or DNA replication were treated to one-cell stage embryos (Figure 4A). When the initiation or elongation of transcription was inhibited by treatment with triptolide (TRP) or 5,6-dichlorobenzimidazole DRB from 4 to 10 hpi (S-phase) after in vitro fertilization (IVF), the R-loop level was significantly reduced in both maternal and paternal pronuclei (Figure 4B). Our quantification showed that the mean value of each inhibitor is 1.0 for Con, 0.69 for DRB, and 0.72 for TRP in the maternal pronucleus, and 1.0 for Con, 0.78 for DRB, and 0.78 for TRP in the paternal pronucleus (Figure 4C). Finally, R-loop levels were dramatically decreased in both pronuclei when the aphidicolin, an inhibitor of DNA replication, was treated.

### 2.5. Anticancer Agents Do Not Alter Zygotic R-Loop Levels Globally

Given that many commercially available anticancer agents such as cisplatin and doxorubicin induce DNA damage, we next investigated whether chemotherapeutic agents that induce DNA damage, such as DNA double-strand breaks (DSBs) [39,40], alter R-loop biogenesis in zygotes. To that end, the DNA damage inducer cis-diamminedichloroplatinum (II) (cisplatin) or doxorubicin was treated to zygotes for 8 h (from 4 hpi to 12 hpi). As shown in Figure 5, although levels of γH2AX, a marker of DNA DSBs, were significantly increased in the cisplatin (Figure 5C) and doxorubicin-treated groups (Figure 5E), no significant changes in R-loop levels were detected between the control and treated groups (Figure 5B,D).

## 3. Discussion

R-loops are focally detected throughout the genome, and their function has been implicated in many molecular processes, including DNA replication, transcription, epigenetic modification, and DNA damage repair [41]. However, no study defining its function during early development has been performed to date.

Given that the S9.6 antibody, which is widely used for the detection of R-loops in IF and immunoprecipitation experiments, often produce unwanted artifacts [42,43], the optimization of the R-loop IF protocol in mouse preimplantation embryos using this antibody was needed first. Studies showed that strong S9.6 signals were detected, especially in cytoplasmic regions, and the signal was not effectively eliminated by treatment with RNase H, which specifically degrades RNA/DNA hybrids [42]. In addition to the DNA:RNA hybrid signals in the nucleus, the majority of cytoplasmic signals recognized by the S9.6 antibody are double-stranded RNA (dsRNA) species [43,44]. Furthermore, it has been shown that although the S9.6 antibody preferentially recognizes GC-rich DNA:RNA hybrid duplexes over dsRNAs and dsDNAs, it still has binding affinity for dsRNAs in vitro [45,46].

Therefore, in this analysis, to reduce cytoplasmic dsRNA signals, HCl-mediated antigen retrieval that was used for the IF of BrdU, 5mC, or 5-hydroxylmethyl cytosine was adopted [47,48,49,50]. The analysis revealed that denaturing double-stranded DNA steps using HCl (antigen retrieval) is required for unmasking R-loops in zygotic pronuclei. Consistently, another study showed that dsRNA can be stably denatured by treatment with HCl [51]. The S9.6 signal was not detected without the denaturation step of nucleic acids, but it was clearly detected in the zygotic pronuclei when treated with 4N HCl protocol (Figure 1A). Using the method, we found that R-loops are distributed genome-wide throughout all of the preimplantation stage embryos we examined. In our analyses, the S9.6 signals were detected not only in the S phase of one-cell stage zygotes, in which transcription and DNA replication occurred, but also at G1 (early PN), G2 (late PN), and metaphase II stage oocytes (Figure 2). The R-loops detected in the non-S phase are possibly associated with the chromosome structure. For example, R-loops are found at centromeres and telomeres, and they participate in chromosome segregation and homologous recombination steps, which are essential for telomere maintenance [52,53]. R-loops were considered byproducts of transcription in early studies [22]. Later, three hypotheses were proposed in the transcription-associated R-loop biogenesis [41]. (1) The first hypothesis is the thread back model, in which in the middle of transcription, a nascent RNA strand synthesized by Pol II invades the DNA duplex, reanneals with template DNA, and displaces a single-strand DNA (non-template) after the nascent RNA exits Pol II [41]. (2) R-loops can also be formed in trans [54]. The nascent RNA can be reannealed with DNA sequences at a different locus if it has a homologous sequence with the template DNA. (3) Lastly, the DNA:RNA hybrid is formed in the transcription machinery and can be extended, leading to R-loop formation [55]. Given that the first transcription in mouse zygotes occurs during the S phase (from late PN3 to PN5) [56], R-loops are detected from PN1 (G1 stage) where no transcription occurs, and R-loop levels do not increase during the S phase, there are possibly unknown mechanisms that involve R-loop formation during the PN. Alternatively, it is also possible that the R-loops detected during the PNs are transmitted from germ cells (oocytes and sperm) or are both newly produced from pronuclei after fertilization and inherited from germ cells. Further studies are required to resolve this issue.

We used DNA replication or transcription inhibitors to investigate which process mainly contributes to R-loop formation in one-cell stage zygotes. Inhibitors of DNA replication (aphidicolin), transcription elongation (DRB), and transcription initiation (triptolide) decreased R-loop levels (Figure 4B) in zygotes at the S phase. In particular, the R-loop levels dramatically decreased when DNA replication was inhibited. There are plausible explanations for our findings. (1) Given that aphidicolin decreases transcription level in one-cell stage zygotes [56], inhibition of DNA replication affects transcription levels. (2) Given that aphidicolin is known to inhibit DNA polymerase alpha and formation of Okazaki fragments [57] and the S9.6 antibodies also bind to the Okazaki fragments [38], the reduction in the S9.6 signal was led by the reduced formation of Okazaki fragments in the presence of aphidicolin. (3) In the S phase, TRCs lead to the formation of DNA double-strand breaks (DSBs) and increased R-loop levels, in which TRCs levels are decreased when DNA replication is inhibited [20]. Our study showed that inhibition of transcription initiation by TRP reduced R-loop biogenesis, whereas one study showed that inhibition of transcription elongation by DRB increased R-loop biogenesis in U-2-OS cells [58]. This might be due to a transcriptionally permissive chromatin state in zygotes, in which transcription can occur without defined core promoter elements [5]. In yeast, it has also been shown that transcription and associated RNA processing steps are the source of DNA damage-causing R-loops, regardless of TRCs [59]. Therefore, the increase in R-loop levels in aphidicolin-treated zygotes is likely, at least in part, due to the aberrant regulation of the transcription machinery.

We also determined R-loop dynamics during one-cell development. The S9.6 signal in the maternal and paternal pronuclei displayed the highest intensities at the PN1 stage. As pronucleus maturation occurs, R-loop levels decrease in both pronuclei, and R-loop levels greatly decrease between the PN1 and PN2 stages (Figure 3A). Interestingly, R-loop levels were constant during the S phase, where transcription and DNA replication occur simultaneously. It is also possible that R-loops detected in early PNs might be related to histone modification and chromatin structure and contribute to genomic reprogramming that occurs in both maternal and paternal genomes during the PNs. In particular, the paternal genome undergoes protamine-to-histone transition from extremely compacted states to loosened states, and the maternal pronucleus has more open chromatin state than other late preimplantation stage zygotes [56]. The specialized chromatin structures of the one-cell stage zygotes cause unique patterns of transcription. For example, zygotic transcription exhibits enhancer-free and core promoter element-independent transcription, namely transcriptionally permissive states. In general, transcripts resulting from minor ZGA contain introns and parts beyond transcription termination sites (TTS), because zygotes have insufficient splicing factors [5]. Therefore, the unique characteristics of minor ZGA may possibly cause the dynamics of R-loop levels and genome-wide distribution in one-cell stage zygotes. Furthermore, in the one-cell stage zygote, DNA DSBs are mainly observed in the PN4-5 stage [60].

We then examined the correlation between R-loop formation and DNA DSBs, and confirmed that DNA damage-inducing agents did not change the R-loop level. This finding differs from previous results in that the R-loop participates in the processes of DNA damage and repair [61]. There are two possible explanations for this finding. First, R-loops related to DNA damage are increased, but there are other factors that rapidly decrease R-loop levels. Cisplatin and doxorubicin are known to directly bind to DNA and inhibit the binding of transcription and replication factors [39,40]. Therefore, they may decrease the transcription and DNA replication-related R-loops. Second, one-cell embryos have unique characteristics in DNA repair mechanisms, such as (1) reliance on maternal mRNAs and proteins, and (2) a lack of cell cycle checkpoints (G1/2 and G2/M) [62]. Therefore, zygotes have a low ability to perform DNA damage repair processes related to R-loop metabolism.

In summary, our study revealed that IF for R-loop detection with the S9.6 antibody in preimplantation embryos requires the HCL-mediated antigen retrieval step. R-loops are widely detected throughout the preimplantation embryo and all cell cycle phases. Next, we found that R-loop levels decreased when pronucleus maturation occurred, especially between the PN1 and PN2 stages. R-loop levels are not affected by an increase in γH2AX levels but are affected by DNA replication and transcription. DNA replication mostly contributed to the formation of the R-loop in the S phase of the one-cell stage zygote. Follow-up studies are necessary to directly evaluate factors affecting minor ZGA in the regulation of R-loop metabolism.

## 4. Materials and Methods

### 4.1. Animals

ICR female (6–8 weeks) and male (8–12 weeks) mice were purchased from Orient Bio Co. Ltd. (Seoul, Korea). Animals were maintained under a 12 h light/dark cycle. Mouse experiments and procedures were performed according to the Konkuk University Guide for the Care and Use of Laboratory Animals (KU22181).

### 4.2. Metaphase II Oocyte Collection and In Vitro Fertilization

Female ICR mice (8–10 weeks) and male ICR mice (8–12 weeks) were used for metaphase II oocytes and collection of spermatozoa, respectively. Superovulation was performed by intraperitoneal (IP) injection of pregnant mare serum gonadotropin (PMSG, 7.5 IU), followed by IP injection of human chorionic gonadotropin (HCG, 7.5 IU) 48 h later for the collection of metaphase II oocytes. Spermatozoa were collected from the caudal epididymis of male ICR mice and capacitated in PVA-mHTF medium containing 0.4 mM methyl-β-cyclodextrin (MBCD) for at least 1 h. After 14–15 h of HCG injection, metaphase II oocytes were collected from the ampullae of the oviducts. The collected oocytes were transferred to mHTF medium containing 0.625 mM GSH and inseminated with capacitated spermatozoa. Two to three hours after IVF, fertilized embryos were washed and cultured in KSOM medium (MR-121-D, EmbryoMax) and fixed at the following time points for each PNs. All the media were pre–incubated for at least 2 h in an incubator (37 °C, 5% CO_2_).

### 4.3. Immunofluorescence and Quantification

Embryos were fixed with 4% paraformaldehyde (PFA)/0.01% PVA-PBS (15 min, room temperature (RT)) at each time point and permeabilized with 0.5% TritonX/0.01% PVA-PBS (20 min, RT). After washing three times with 0.05% Tween 20/0.01% PVA-PBS (10 min, RT), embryos were denatured in 4N HCl (15 min, RT) and neutralized in 100 mM Tris-HCL (pH 8.5) for 20 min at RT. After washing several times, the embryos were blocked in 5%BSA/0.01% PVA-PBS (2 h, RT) and incubated with each primary antibody (anti-S9.6 (1:200, MABE1095, Merck, Rahway, NJ, USA) for detecting R-loop, anti-γH2AX (1:400, 2577, Cell Signaling, Danvers, MA, USA) for detecting DNA damage, and anti-H3K4me3 (1:200, 39160, Active Motif, Carlsbad, CA, USA) for detecting both male and female pronuclei) overnight at 4 °C, followed by incubation with secondary antibodies (Alexa Fluor 488 (1:400) and Alexa Fluor 546 (1:400)) for 1 h at RT. After washing, DNA was counter-stained with 2 uM To-PRO3 Iodide (T3605, Thermo Fisher Scientific, Carlsbad, CA, USA) and then mounted with H-1000 (Vectashield w/o DAPI) or H-1200 (Vectashield with DAPI). The mounted embryos were imaged using a laser confocal microscope (LSM 8000, Zeiss, Oberkochen, Germany) at 40× or 63× magnification. Single optical sections of the middle of the maternal and paternal nuclei are presented as representative images and used for the quantification of immunofluorescence intensity using the ImageJ program. The following quantification method was used to measure S9.6 intensity: total intensity of pronucleus = (mean intensity of the pronucleus region × area of pronucleus)−(mean intensity of cytoplasmic region × area of the pronucleus).

### 4.4. Treatment of Inhibitors

To inhibit DNA replication and transcription in 1-cell stage zygotes, zygotes were transferred to KSOM containing 100 μM DRB (D1916, Sigma, Livonia, MI, USA), 1 μM triptolide (T3652, Sigma, Livonia, MI, USA), or 2 μg/mL aphidicolin (84958, Sigma, Livonia, MI, USA) between 4 hpi and 10 hpi. To induce DNA damage, zygotes were transferred to KSOM containing 5 µM cisplatin (P4394, Sigma, Livonia, MI, USA) and 200 nM doxorubicin (D1515, Sigma, Livonia, MI, USA) at 4 hpi and 12 hpi. After several washing steps, inhibitor-treated zygotes were fixed with 4% PFA and subjected to immunofluorescence analysis. Control embryos were cultured in inhibitor-free KSOM medium containing the appropriate amount of dimethyl sulfoxide (DMSO), a solvent used for the preparation of inhibitors.

### 4.5. Rnase H Treatment

After denaturation/neutralization steps, embryos were washed several times and incubated with 80 U/mL Rnase H (M2097, NEB, Ipswich, MA, USA) in Rnase H buffer for 3 h at 37 °C to confirm that the S9.6 signal is from RNA/DNA hybrid. After several washings, embryos were subjected to immunofluorescence.

### 4.6. Statistical Analyses

Unpaired *t*-test was used to determine differences in pronuclear intensities of IF with S9.6 antibody between two groups, and one-way analysis of variance (ANOVA) was used to determine differences in pronuclear intensities of IF with S9.6 antibody among ≥3 groups, followed by Tukey’s test or Kruskal–Wallis post hoc test to determine significant differences between pairs. *p*-values less than 0.05 were considered statistically significant. At least three biological replicates were used to determine the statistical significance of each experiment.

## Figures and Tables

**Figure 1 ijms-23-14345-f001:**
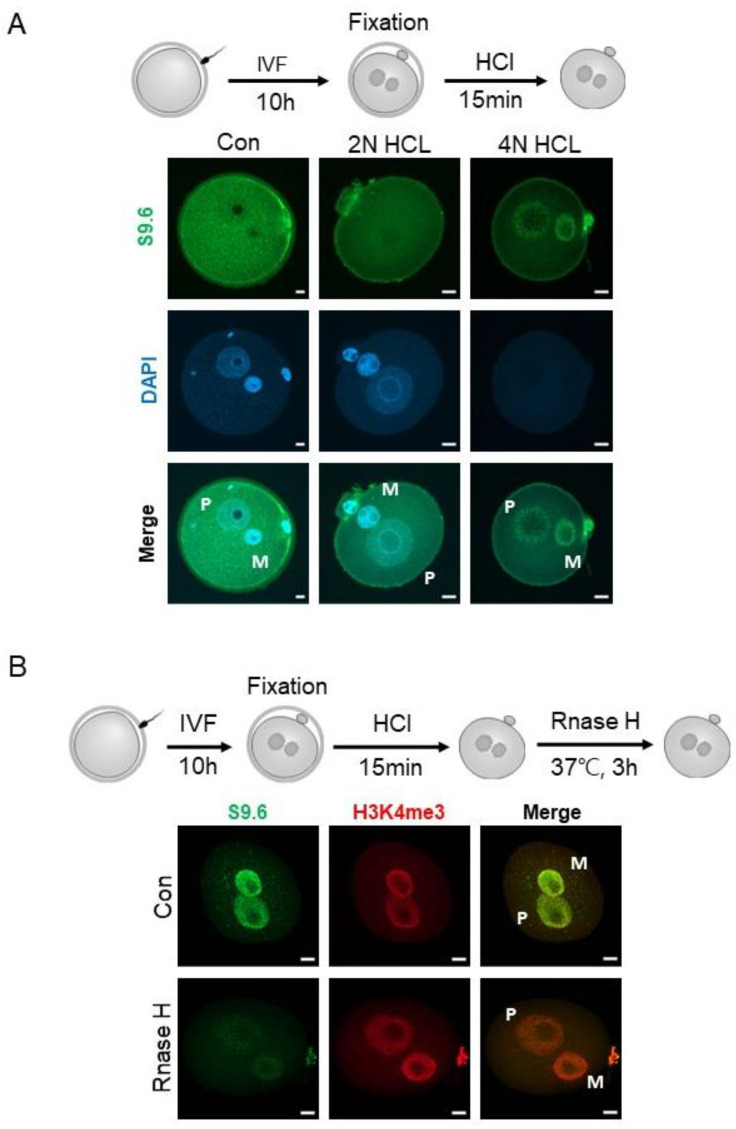
Optimization of R-loop immunofluorescence. (**A**) Maximum intensity projection of z-stack images of 10 hpi zygotes. Immunofluorescence detection of the R-loop using S9.6 antibodies. DNA denaturation steps were used for antigen retrieval using the 2N or 4N HCl for 15 min. Images show the R-loop (S9.6 antibodies, green), DNA (DAPI, blue), and merged image of these two channels (merge). (**B**) Maximum intensity z-stack projection images of 10 hpi zygotes. Immunofluorescence detection of the R-loop (S9.6 antibodies, green), histone H3K4me3 (red), DNA (DAPI, blue), and merged images of these three channels (merge) in zygotes that were mock-treated or treated with 80 IU/mL of RNase H for 3 h at 37 °C. M, maternal pronucleus; P, paternal pronucleus. All scale bars = 10 µM.

**Figure 2 ijms-23-14345-f002:**
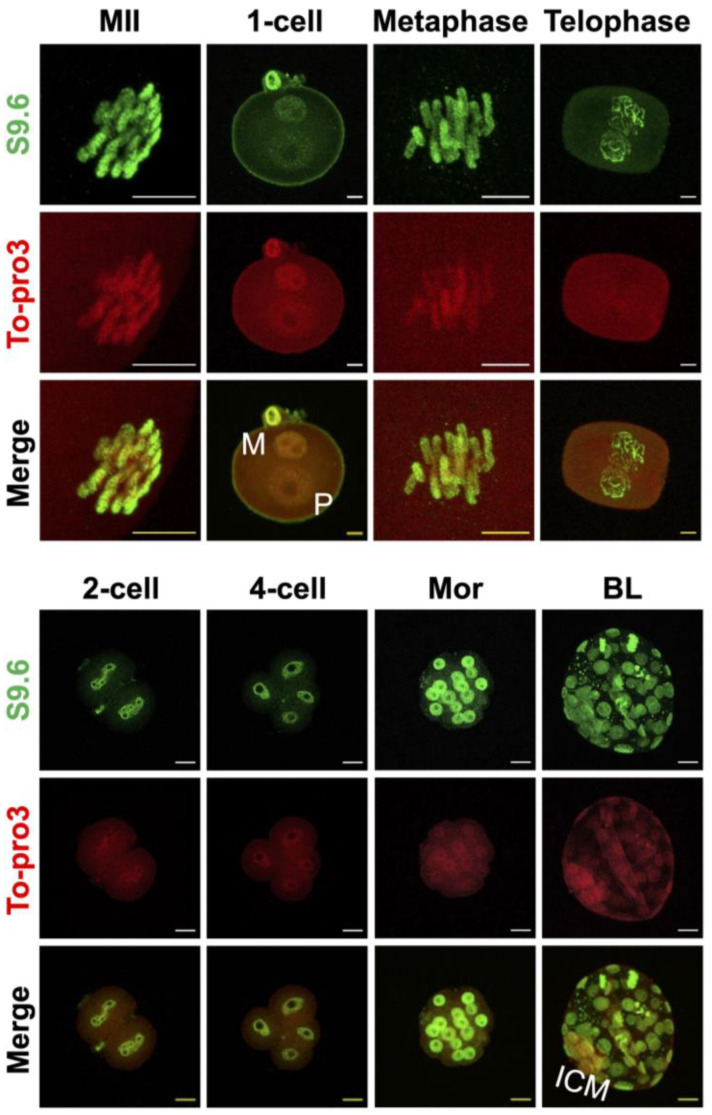
R-loops persist through the pre–implantation stage embryo. Representative images of the maximum intensity z-stack projection of each preimplantation stage embryo. Immunofluorescence detection of the R-loop (S9.6 antibodies, green), DNA (To-pro3, red), and merged image of these two channels (merged) in metaphase II stage oocytes (MII), pronucleus stage zygote (1-cell), metaphase stage of the 1-cell zygote (metaphase), telophase stage of the 1-cell zygote (telophase), 2-cell, 4-cell, Mor (morula), and BL (blastocyst). M, paternal pronucleus; P, paternal pronucleus; ICM, inner cell mass. All scale bars = 10 µM.

**Figure 3 ijms-23-14345-f003:**
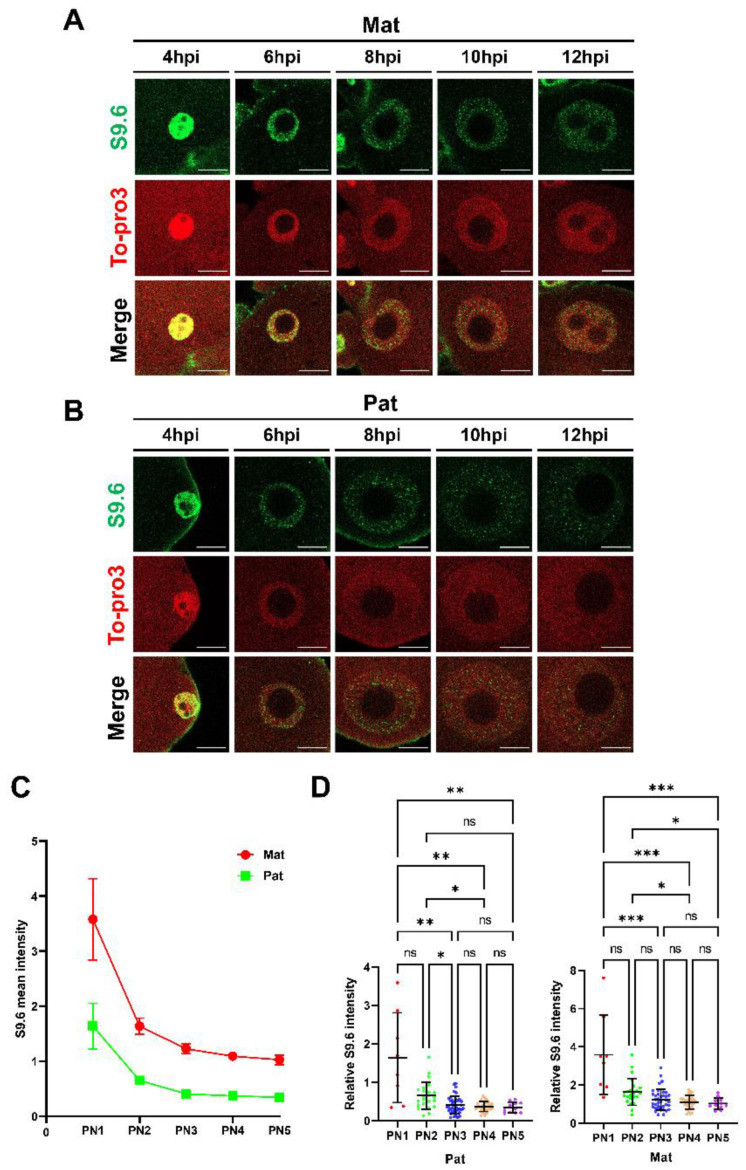
Dynamic R-loop levels through pronuclear stages (PNs). (**A**) Single confocal optical section images of each maternal pronucleus stage. Immunofluorescence detection of the R-loop (S9.6 antibodies, green), DNA (To-pro3, red), and merged images of these two channels (merged). (**B**) Single confocal optical section images of each paternal pronucleus stage. Immunofluorescence detection of the R-loop (S9.6 antibodies, green), DNA (To-pro3, red), and merged images of these two channels (merged). All scale bars = 10 µM. (**C**) Comparison of relative S9.6 signal intensities in maternal (red) and paternal (green) pronucleus from PN1 to PN5 stages. Each plot indicates the mean values of the relative S9.6 signal intensities of each PN. (**D**) Quantification of relative S9.6 signal intensities in the maternal (**left**) and paternal (**right**) pronuclei. PN1; n = 8, PN2; n = 23, PN3; n = 38, PN4; n = 28, PN5; n = 11. The experiment was independently replicated three times. Statistical analysis was performed using one-way ANOVA with Tukey’s test and Kruskal–Wallis test. M, paternal pronucleus; P, paternal pronucleus. Error bars represent standard deviation. *p*-value: * *p* < 0.05, ** *p* < 0.01, *** *p* < 0.001; center values, mean; ns, not significant.

**Figure 4 ijms-23-14345-f004:**
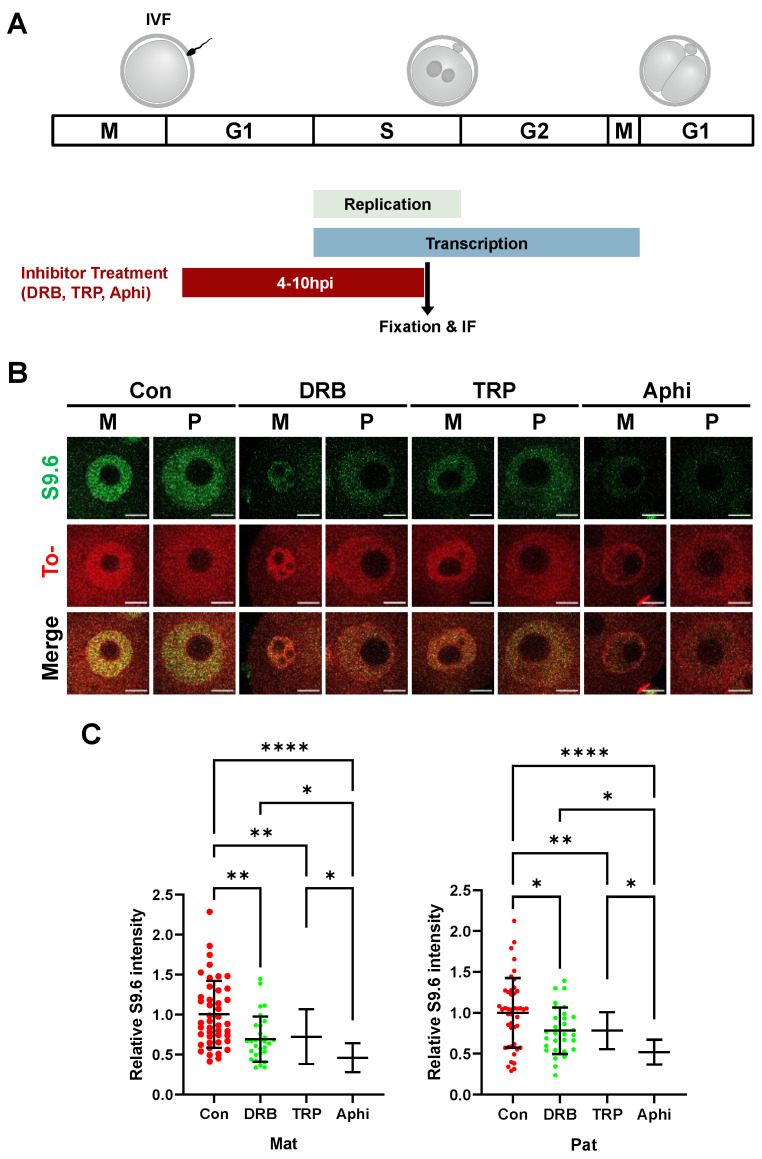
Zygotic R-loop levels are affected by DNA replication and transcription. (**A**) Schematic diagram of experimental design for the inhibitor treatment of transcription or DNA replication from 4 to 10 hpi (S-phase) after IVF. ZGA, zygotic genome activation; hpi, hours post–insemination; IF, immunofluorescence. (**B**) Single confocal optical section images of the maternal and paternal pronuclei. Zygotes were mock-treated (Con) and treated with 100 µM 5,6-dichloro-1-β-D-ribofuranosyl-benzimidazole (DRB), 1 µM triptolide (TRP), and 2 µg/mL aphidicolin (Aphi) from 4 hpi to 10 hpi (S-phase). Immunofluorescence detection of the R-loop (S9.6 antibodies, green), DNA (To-pro3, red), and merged images of these two channels (merged). All scale bars = 10 µM. (**C**) Quantification of relative S9.6 signal intensities in each maternal (**left**) and paternal (**right**) pronuclei. Con; n = 44, DRB; n = 31, TRP; n = 41, Aphi; n = 20. This experiment was independently replicated three times. Statistical analysis was performed using one-way ANOVA with the Kruskal-Wallis test. Error bars represent standard deviation. M, paternal pronucleus; P, paternal pronucleus. *p*-value: * *p* < 0.05, ** *p* < 0.01, **** *p* < 0.0001; center values, mean.

**Figure 5 ijms-23-14345-f005:**
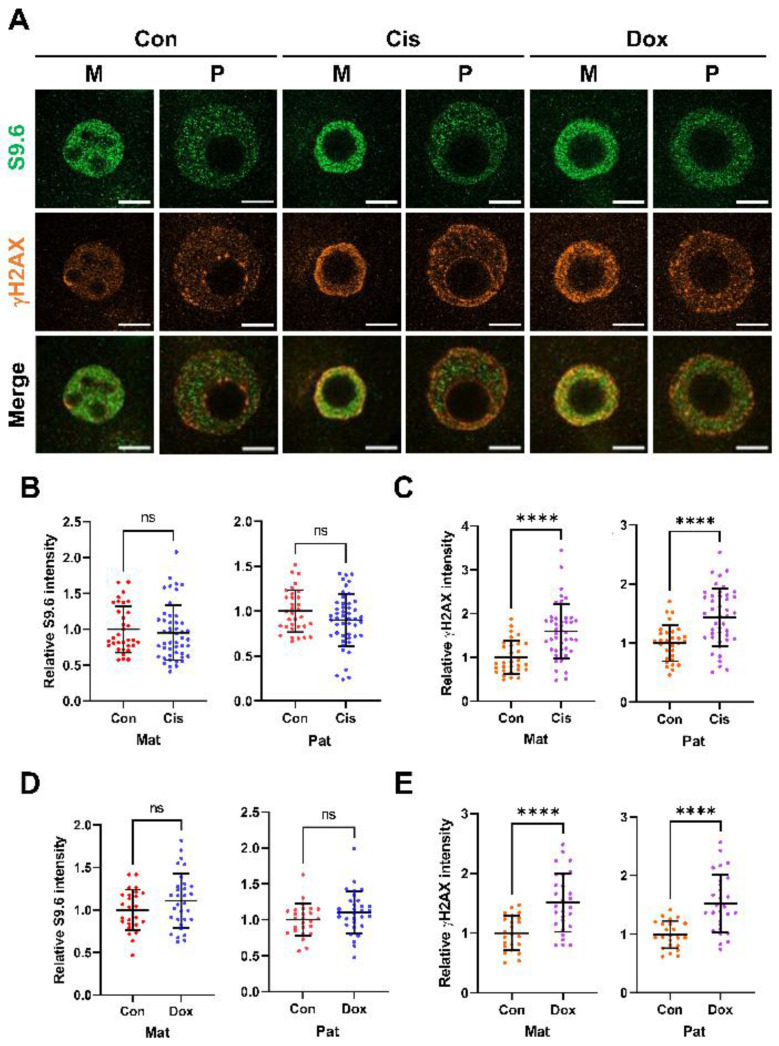
Increased γH2AX levels did not affect the R-loop levels. (**A**) Single confocal optical section images of the maternal (M) and paternal (P) pronuclei from representative embryos. Zygotes were mock-treated (Con) and treated with cisplatin (Cis) and doxorubicin (Dox) from 4 hpi to 12 hpi (S-G2 phase). Immunofluorescence detection of the R-loop (S9.6 antibodies, green), γH2AX (yellow), and merged image of these two channels (merged). M, paternal pronucleus; P, paternal pronucleus. All scale bars = 10 µM. (**B**) Quantification of relative S9.6 signal intensities in each maternal (Mat, **left**) and paternal (Pat, **right**) pronuclei. Con; n = 33, Cis; n = 48. (**C**) Quantification of relative γH2AX signal intensities in maternal (Mat, **left**) and paternal (Pat, **right**) pronuclei. Con; n = 30, Cis; n = 43. (**D**) Quantification of relative S9.6 signal intensities in each maternal (Mat, **left**) and paternal (Pat, **right**) pronuclei. Con; n = 26, Dox; n = 48. (**E**) Quantification of relative γH2AX signal intensities in maternal (Mat, **left**) and paternal (Pat, **right**) pronuclei. Con; n = 23, Cis; n = 28. This experiment was independently replicated three times. Statistical analysis was performed using an unpaired t-test. Error bars represent standard deviation. *p*-value: **** *p* < 0.0001; center values, mean; ns, not significant.

## Data Availability

Not applicable.

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
