# Peer review of "Dynamic Change of R-Loop Implicates in the Regulation of Zygotic Genome Activation in Mouse"

_ijms, 2022, doi:10.3390/ijms232214345_

Round 1

Author Response

The present study is interesting, actual  and important, since it brings methodology for the study and new informations about the formation of R-loops during the zygote formation and  initial period of embryonic development of mammals, more precisely, the cleavage period, which leads to the formation of the blastula, also called blastocyst. In this sense, I do not see any methodological, interpretation or writing problems with the article.

Response: The authors thank the reviewer very much for his/her insightful comments.

Minor Comments:

  1. In all figures, the value of the length of the scale bars in micrometers is indicated with μM, but the correct is μm. This needs to be corrected.

Response: We have corrected it in the revision.

  1. The first genetic programming takes place in the morula stage, which determines that the outer cells of the morula will become the blastocyst wall (the trophoblast), while the inner cells become the inner embryonic mass (embryoblast). Such programming determines which trophoblast cells will originate one of the components of the chorion (chorion epithelium or cyto and syncyciotrophoblast, depending on the species of mammal in question) and that the cells of the inner embryonic mass will follow the path that will lead to the formation of the embryo, yolk cavity , amniotic cavity, allantois and connective tissue and chorion vessels. In the Results, item 2.2., it is not clear whether R-loops are present in the external and internal cells of the morula, nor if they are present in the trophoblast and embryoblast. It is important that authors provide such information. The photos in Figure 2, referring to the morula and blastocyst phases, apparently show marking on the outer cells and trophoblast. Is there or not labeling in the inner cells and in the cells of the embryoblast, respectively??

Response: The authors thank you so much for the intriguing comment. Based on our immunofluorescence, we do not see any differences of S9.6 IF intensities between external and internal cells of the morula stage embryos, and between trophoblast and ICM of the blastocyst stage embryos. As the reviewer suggested, we have added the following sentences in the result section. “The R-loop IF signals were also evident in both external and internal cells of the morula, and both trophoblast and inner cell mass of the blastocyst at similar levels.” in line 166.

Given my long experience in teaching embryology, I will allow myself to make some observations and suggestions for future studies to be carried out by the authors involving the early embryonic development of mammals.

  1. It is important to remember that the zygote is nothing more than the oocyte II after fertilization and completion of meiosis, followed by the junction of male and female pronuclei. In this sense, transcripts of maternal origin will always exist, given that half of the genome has this origin. From the formation of the zygote, it is very important to know if the genes transcribed during the first 3 zygotic mitotic divisions (phases of 2, 4 and 8 blastomeres) are of maternal or paternal origin, which types of proteins they encode and which of them are more susceptible to the formation of R-loops, if R- loops are more frequent on chromosomes of maternal or parental origin. Of course, there is a need to establish whether they are dominant or recessive genes, in order to better understand phenotypic formation and even embryonic viability at this stage

Response: The authors greatly appreciate the reviewer’s insightful suggestion. We agree with the reviewer’s idea. Further studies will answer the reviewer question that has been posed in the field of developmental biology.

  1. It is very important when designing new experiments to remember that important cellular and genetic processes take place during the cleavage period. Until the 8-blastomere stage, before compaction, any blastomere that loses contact with the others starts to act as a zygote, continues the cleavage process and gives rise to a blastocyst. This suggests the existence of similarity of transcripts between blastomeres and between them and the zygote, given that they act as a zygote if they separate from the others, making the blastomeres the main and most important embryonic stem cells.. Not least, it is important to remember more one time that the first genetic programming takes place in the morula stage, which determines that the outer cells of the morula will become the blastocyst wall (the trophoblast), while the inner cells become the inner embryonic mass (embryoblast). Such programming determines which trophoblast cells will originate one of the components of the chorion (chorion epithelium or cyto and syncyciotrophoblast, depending on the species of mammal in question) and that the cells of the inner embryonic mass will follow the path that will lead to the formation of the embryo, yolk cavity , amniotic cavity, allantois and connective tissue and chorion vessels. That said, in my opinion, studies analyzing which genes are activated or blocked in the external and internal cells of the morula, whether the transcribed or blocked genes are of maternal or paternal origin, what types of proteins they encode and which of them are more susceptible to the formation of R-loops, if R-loops are more frequent in chromosomes of maternal or parental origin, will significantly contribute to the understanding of genetic programming and reprogramming involved in the genetic regulation of mammalian embryonic development during the cleavage period.

Response: The authors thank the reviewer for the insightful comment. We will pursue some experiments to determine precise roles of R-loop in the genetic programming and reprogramming of preimplantation embryos.

Reviewer 2 Report

The authors have presented a very interesting research. However, its explanation and scientific designing way of research have needed more attention. So, some suggestions/ corrections are required before considering for publication.

o   The language should be revised because some sentences are hard to understand.

o   Abbreviations must be defined and kept consistency throughout the article.

o   Try new recent five years published papers should use for citation.

o   Why author did chose cauda part of epididymis, instead of other parts, Please clarify it this part is more important as compare to others with cited references?

o   In result section avoid the again and again usage of word “we”.

o   In conclusion, this text appears to be a sort of catalogue of information and needs a serious revision of English stylization. For better readability try the author to render it more fluid.

Author Response

The authors have presented a very interesting research. However, its explanation and scientific designing way of research have needed more attention. So, some suggestions/ corrections are required before considering for publication.

Response: The authors thank the reviewer very much for his/her insightful comments.

  1. The language should be revised because some sentences are hard to understand.

Response: We have changed the unclear sentences in the revision.

  1. Abbreviations must be defined and kept consistency throughout the article.

Response: We have corrected them in the revision.

  1. Try new recent five years published papers should use for citation.

Response: We have revised and tried to cite both historical and recent references as many as we can in the revision. 

  1. Why author did chose cauda part of epididymis, instead of other parts, Please clarify it this part is more important as compare to others with cited references?

Response: Sperm from the caudal epididymis are routinely used for the purpose of IVF as sperm in the caudal epididymis are mature enough for fertilization just like ejaculated ones.

  1. In result section avoid the again and again usage of word “we”.

Response: We have revised and minimized the use of word “we” in the revision.

In conclusion, this text appears to be a sort of catalogue of information and needs a serious revision of English stylization. For better readability try the author to render it more fluid.

Response: The authors thank the reviewer for giving us a chance to significantly improve our manuscript.

Reviewer 3 Report

            This manuscript provides some interesting data on the prevelance of R-Loops throughout early mouse embryogenesis.  The data are convincing, particularly the Rnase experiment in Fig. 1B.  However, because virtually all the data I based on immunocytochemistry, this manuscript would be more convincing if a few additional questions were answered.  The main issue is why the embryo needs 4 N HCl denaturation to reveal the R-Loop while in other cells this seems not to be required.  It is unclear why embryonic cells need this additional treatment if the antibody identifies R-Loops without HCl denaturation in other cell types.    If 4 N NCl denatures DNA to reveal the antigen, does it also denature the DNA/RNA hybridization?  Two experiments would help with this.

            First, the experiment in Fig. 2 of the embryos at different stages of development could be repeated with an Rnase control, just as in Fig. 1.  This would lend credence to the conclusion that R-Loops are really present.  

            Second, a control cell, possibly ES cells, in which R-Loops have been shown with this antibody without 4 N N HCl, could be tested with and without the HCl to verify that it has no effect on the signal.  

Fig. 4 is a little confusing.  The embryos were treated with inhibitors, then they were fixed, but when were they fixed? It must have been in the 1Cell stage since the embryos still have PN, but the diagram in Fig. 4A continues to the 2Cell stage.  Please clarify. 

Author Response

Dear Reviewer 3,

Attached please see our point-by-point responses to your comments. Thank you. 

Reviewer 4 Report

Dynamic change of R-loop implicates in the regulation of zygotic genome activation in mouse

By Hyeonji Lee, Seong-Yeob You, Dong Wook Han, Hyeonwoo La, Chanhyeok Park, Seonho Yoo, Kiye Kang, Min-Hee Kang, Youngsok Choi, Kwonho Hong

The article studies change in R-loop dynamics during mouse zygotic development

Data are very interesting and novelty.

I have some concerns about the article.

Introduction

Page 2, line 73. Please indicate the main roles of R-loop  in cells.

Materials and Methods

Page 7, line 217 and page 12, line 434. The correct term should be in vitro fertilization, as is included in figures 1 and 4. Insemination is used for placing semen in intrauterine cavity. This is not the case.

Page 12, lines 447-448. It should be indicated why to detect S9.6, 447 γH2AX and H3K4me3 and their role in R-loop dynamics.

Statistical analysis

What variables were analyzed by unpaired t-test and what variables were analyzed by ANOVA? These should be clarified

Author Response

The article studies change in R-loop dynamics during mouse zygotic development. Data are very interesting and novelty. I have some concerns about the article.

Response: The authors thank the reviewer very much for his/her support and insightful comments.

  1. Introduction: Page 2, line 73. Please indicate the main roles of R-loop in cells.

Response: Indeed, the main roles of R-loop are described in the line 76-80.

  1. Materials and Methods: Page 7, line 217 and page 12, line 434. The correct term should be in vitro fertilization, as is included in figures 1 and 4. Insemination is used for placing semen in intrauterine cavity. This is not the case.

Response: We have corrected it in the revision.

  1. Page 12, lines 447-448. It should be indicated why to detect S9.6, 447 γH2AX and H3K4me3 and their role in R-loop dynamics.

Response: We have added purpose of the antibodies in the revision.

  1. Statistical analysis: What variables were analyzed by unpaired t-test and what variables were analyzed by ANOVA? These should be clarified

Response: We have added more information of the “variables” in the revision.

Round 2

Reviewer 2 Report

I recommended this paper publishing on IJMS, because the authors are revised the manuscript carefully.

Reviewer 3 Report

The authors have addressed my concerns